

# A model based LSTM and graph convolutional network for stock trend prediction

Xiangdong Ran[1], Zhiguang Shan[2], Yukang Fan[3] and Lei Gao[4]

[1] Beijing Information Technology College, Beijing, China
[2] Informatization and Industry Research Department, State Information Center, Beijing, China
[3] College of Arts and Science, New York University, New York, United States of America
[4] Standards and Safety Department, Beijing Big Data Center, Beijing, China

## ABSTRACT

Stock market is a complex system characterized by collective activity, where inter-dependencies between stocks have a significant influence on stock price trends. It is widely believed that modeling these dependencies can improve the accuracy of stock trend prediction and enable investors to earn more stable profits. However, these dependencies are not directly observable and need to be analyzed from stock data. In this paper, we propose a model based on Long short-term memory (LSTM) and graph convolutional network to capture these dependencies for stock trend prediction. Specifically, an LSTM is employed to extract the stock features, with all hidden state outputs utilized to construct the graph nodes. Subsequently, Pearson correlation coefficient is used to organize the stock features into a graph structure. Finally, a graph convolutional network is applied to extract the relevant features for accurate stock trend prediction. Experiments based on China A50 stocks demonstrate that our proposed model outperforms baseline methods in terms of prediction performance and trading backtest returns. In trading backtest, we have identified a set of effective trading strategies as part of the trading plan. Based on China A50 stocks, our proposed model shows promising results in generating desirable returns during both upward and downward channels of the stock market. The proposed model has proven beneficial for investors to seeking optimal timing and pricing when dealing with shares.

## INTRODUCTION

Stock market is a complex system characterized by collective activity, where stocks are increasingly correlated and influence each other in generating volatility and comovement, driven by factors such as the trading of exchange-traded funds (ETFs) and financialization characterized by the presence of institutional investors.

ETFs, which track the prices of a basket of stocks, are increasingly favored by both retail and institutional investors. This success is well-deserved, as ETFs offer unprecedented diversity at a low cost and high liquidity. The big investors in the asset management

Corresponding author
Xiangdong Ran,
ranxiangdong@hotmail.com

industry are increasingly allocating their investments to ETFs. Exchange traded products accounted for approximately 40% of all trading volume in US markets as early as in August 2010 (*BlackRock, 2011*). The potential for ETFs to transmit their demand shocks to their underlying stocks is significant, such stock volatility and turnover (*Itzhak, Francesco & Rabih, 2018*; *Hou et al., 2021*). When ETFs are traded, all their underlying stocks are traded simultaneously, causing a liquidity shocks to their underlying stocks. Stocks with higher ETF ownership tend to exhibit higher volatility and turnover, leading to a common fluctuations in their prices.

Arbitrage activities can contribute to an increased volatility in the stock market. There are two ways for arbitrage activities between ETFs and their underlying securities. In high frequency trades, arbitrageurs take long and short positions in ETFs and their underlying baskets and wait for the convergence of prices. In lower frequency trades, arbitrageurs engage in the creation and redemption of ETFs to profit from mispricing. Liquidity shocks in the ETFs market are propagated *via* arbitrage trades to the prices of the underlying securities, adding a new non-fundamental volatility. Whenever the cost of arbitrage is higher, as evidenced by a higher bid-ask spreads or an increased stock borrowing and lending costs, the arbitrage impact is reduced.

Index trading contributes to a generation of non-fundamental volatility and comovement (*Basak & Pavlova, 2015*; *Basak & Pavlova, 2013*). Financialization increase the correlations of stocks, particularly for index futures compared to for non-index ones. Most professional portfolio managers are specialized in a couple of strategies and these strategies often involve a similar set of stocks. For example, value investing tilts to firms with high earning-to-price ratio, while momentum strategy focuses on firms with higher returns during the past year (*Basak & Pavlova, 2015*). Institutions optimally tilt their portfolios towards stocks that comprise their benchmark index. The resulting price pressure boosts index stocks. By demanding a higher fraction of risky stocks than retail investors, institutions amplify the index stock volatilities and aggregate stock market volatility. Trades by institutions induce excess correlations among stocks that belong to their benchmark index (*Basak & Pavlova, 2013*).

Considering stock markets data for stock trend prediction is a challenging but equally rewarding task in financial time series prediction (*Abu-Mostafa & Atiya, 1996*). Traditional machine learning methods are usually quite popular as the preferred choices (*Menon et al., 2016*; *Parray et al., 2020*). In recent years, deep learning methods have performed better than shallow counterparts due to the ability of effective features extract. Among the various deep learning methods, LSTM network (*Hochreiter & Schmidhuber, 1997*) can effectively capture the time series features of stocks and achieves great success in stock price predicting. Graph convolutional network (GCN) can operate directly on graphs and consider the relationship between operational objects (*Kipf & Welling, 2017*).

It is believed that modeling *via* deep learning methods the correlation benefits the task of stock trend prediction to help investors earn more stable profits, such as the use of LSTM and GCN in *Zhao et al. (2023)*. In *Zhao et al. (2023)*, all outputs of the hidden layers of LSTM were not directly utilized as the timing sequence features of the stock instead of the last output of LSTM, which is insufficient for feature extraction. The stock relationship

only qualitative by using a K-means model to return an optional value 0 or 1, which is insufficient for quantification. In this paper, we propose an LSTM-based and graph convolutional network model for stock trend prediction., which includes correlations and influences on stock prices. The proposed model utilizes an LSTM to extract the feature matrices from the corresponding stock, and all outputs of the LSTM's hidden state are used to construct the graph node. Pearson correlation coefficient is employed to integrate the feature matrices in order to generate the adjacency graph matrix that represents the stock networks. The main contributions of this paper may be summarized as follows.

- We proposed a based LSTM and graph convolutional network for stock trend predict. In which, the feature matrices are extracted from the corresponding stock information by the LSTM. All outputs of hidden state of LSTM network were utilized as the graph node embedding of stock. The stock networks are constructed by Person correlation coefficient creating the adjacency graph matrix using the feature matrices.
- We compare the proposed model with several trend prediction methods in prediction performance based on China stocks A50 and the proposed sliding window method. The proposed model outperforms all baseline methods in Acc, $Pre_{pos}$ and $Pre_{neg}$ metrics, remains medium values of $Rec_{pos}$, $Rec_{neg}$, $F1_{pos}$ and $F1_{neg}$ metrics. The visualization demonstrates that information from the input data can be effectively extracted and centrally recorded.
- We identify a set of trading strategies as part of the trading plan to make more money than they lose in the stock trading backtest based on China stocks A50. The stock trading backtest demonstrate that the proposed model can gain better returns regardless of the upward or downward range of the stock index, has a good stability and robustness than the comparative methods.

The rest of this paper is organized as follows. 'Related Work' reviews the related works. 'Methods' describes the proposed model for stock trend prediction. 'Experiments' outline the experimental settings. 'Results' show and analyze the comparison results of different methods. 'Conclusion' presents the conclusion and future work.

## RELATED WORK

### Traditional machine learning techniques

Traditional machine learning methods are widely used for stock trend prediction, such as K-nearest neighbor (KNN), Random Forest (RF), and support vector machine (SVM). Specifically, SVM, RF, and combined approaches are highly favored.

Impact of many factors on stock price makes the stock price prediction a difficult and highly complicated task. In order to overcome such difficulties, *Kumar et al. (2018)* applied machine learning techniques for the stock price prediction *i.e.,* SVM, RF, KNN, Naive Bayes, and Softmax, and applied several technical indicators to stock prices data. *Menon et al. (2016)* opted for a basic ARMA-type predictor for their bulk price prediction. *Kara, Boyacioglu & Baykan (2011)* used artificial neural network (ANN) and SVM to predict the movement direction of Istanbul Stock Exchange National 100 Index. ANN predicts the index significantly better than SVM (*Kara, Boyacioglu & Baykan, 2011*). In *Zhang, Li*

*& Pan (2016)*, a status box method is proposed. Then an ensemble method integrated with AdaBoost algorithm and probabilistic multi-class SVM is constructed to perform the status boxes classification for stock trend prediction. A weighted kernel least squares SVM method was used to predict stock trend in *Markovic et al. (2017)*. The feature ranking and selection were finished through the analytic hierarchy process providing feature weights. In *Özorhan, Toroslu & Şehitoğlu (2019)*, the authors proposed a novel approach based on a modified version of the Zigzag technical indicator, expectation maximization, and SVM for predicting short-term trends in the financial time series of foreign exchange market. In *Parray et al. (2020)*, SVM, perceptron, and logistic regression were used to predict the trend of stocks at next day. In *Picasso et al. (2019)*, the authors combine the technical and fundamental analysts approaches to market trend forecasting through the use of machine learning techniques applied to time series prediction and sentiment analysis in the NASDAQ100 index.

## Deep learning methods

With increasing computational intelligence capacity, deep learning methods have been widely used to predict stock trend in recent years. Deep learning methods perform well and outperform shallow counterparts, due to their ability to effectively extract high-level features from stock data. LSTM was introduced by *Hochreiter & Schmidhuber (1997)* aiming to achieve a better performance by tackling the vanishing gradient issue that recurrent networks would suffer when dealing with long data sequences. LSTM can effectively extract the time series features and achieves better success.

In *Fischer & Krauss (2017)*, the authors made an earlier attempt to apply LSTM network to predict out-of-sample directional movements for the constituent stocks on the S&P 500. The experimental results demonstrate that LSTM network outperforms memory free classification methods, *i.e.,* RF, deep neural net, and logistic regression classifier. In *Wu et al. (2020)*, a novel data annotation method was proposed to extract continuous trend features from financial time series data. The extracted trend features were used for four supervised machine learning methods and two deep learning models for financial time series forecasting. In *Wu et al. (2020)*, a new approach to data annotation was employed to capture the continuous trend characteristic of financial time series data. These features were then utilized in four different supervised machine learning techniques and two deep learning models for predicting financial time series. In *Nelson, Pereira & de Oliveira (2017)*, the authors studied the use of LSTM network to predict future trends of stock prices based on stock market price history and technical analysis indicators. In *Qin et al. (2017)*, the authors propose a dual-stage attention-based recurrent neural network. The proposed method consists of an encoder with an input attention mechanism for adaptively selecting the most relevant input features and a decoder with a temporal attention for appropriately capturing the long-term temporal dependencies of the time series. In *Zhao et al. (2021)*, three prediction models were developed to predict stock trends using the attention mechanism on RNN, LSTM, and gated recurrent unit. In *Kim & Won (2018)*, the authors proposed a new hybrid LSTM network to predict stock price volatility that combines the LSTM network with various generalized autoregressive conditional

heteroscedasticity-type models. In *Chen & Ge (2019)*, the authors explore the attention mechanism in LSTM network based stock price movement prediction. The evaluation results show the effectiveness of the proposed method in the Hong Kong stock market.

The latest deep learning framework, Transformer, has recently been applied to stock prediction (*Wang et al., 2022*; *Lai et al., 2023*; *Zhang et al., 2022*; *Zeng et al., 2023*), because it can better characterize the underlying rules of stock market dynamics by using the encoder–decoder architecture and the multi-head attention mechanism. In *Wang et al. (2022)*, a Transformer is utilized to predict major stock market indices worldwide. A moving window approach is utilized to construct features and labels from the observed time series. Unlike the original decoder in *Vaswani et al. (2017)*, the Transformer do not use the mask attention mechanism. In *Zeng et al. (2023)*, the authors propose a method to forecast whether the price will go up, down or remain the same in the future, employing CNNs to model short-term dependencies and Transformers to learn long-term dependencies within a time series.

## Graph-based approach

In recent years, many studies have shown that considering the correlation among stocks can further improve the prediction performance of stock trend.

*Kipf & Welling (2017)* proposed an efficient variant of convolutional neural networks for semi-supervised classification on graph-structured data. In a number of experiments on citation networks and on a knowledge graph dataset, the proposed approach outperforms the related methods by a significant margin. In *Chen et al. (2020)*, the authors propose a graph convolutional feature based convolutional neural network for stock trend prediction, in which an improved graph convolutional network was used to extract stock market features from the overall stock market information and a Dual-CNN was used to extract individual stock features from specific individual stock information. In *Li et al. (2022a)*, to solve the quantification difficulty problem of the technical charts of stock movement, an arbitrary graph kernel and graph convolutional network are proposed to mine the information in the technical charts for stock movement prediction after extracting the key point sequence from the stock price series. In *Zhao et al. (2022)*, the authors proposed a self-generating relations algorithm based on feature threshold and Hamming Distance to extract relational features automatically from related stocks data. The relational features are used as the adjacency matrix to an graph convolutional neural network. In *Zhao et al. (2023)*, the authors utilized a time series relational model (TSRM) to automatically classify stocks through a K-means model and derives stock relationships, such as industry and upstream/downstream connections. The authors used an LSTM network to extract the time series information and a Graph convolutional network to extract relationship information. In *Li et al. (2020)*, to leverage the connection among stocks, the authors adopted relational graph convolutional networks (RGCN) (*Schlichtkrull et al., 2018*) to encode the graph structure with their correlation matrix. In the correlation matrix, two kinds of relationship representing positive and negative correlation . The authors proposed to add an LSTM between RGCN layers so that the LSTM can dynamically select which part of the information should be transmitted to upper layers. In *Li et al. (2022b)*, the authors

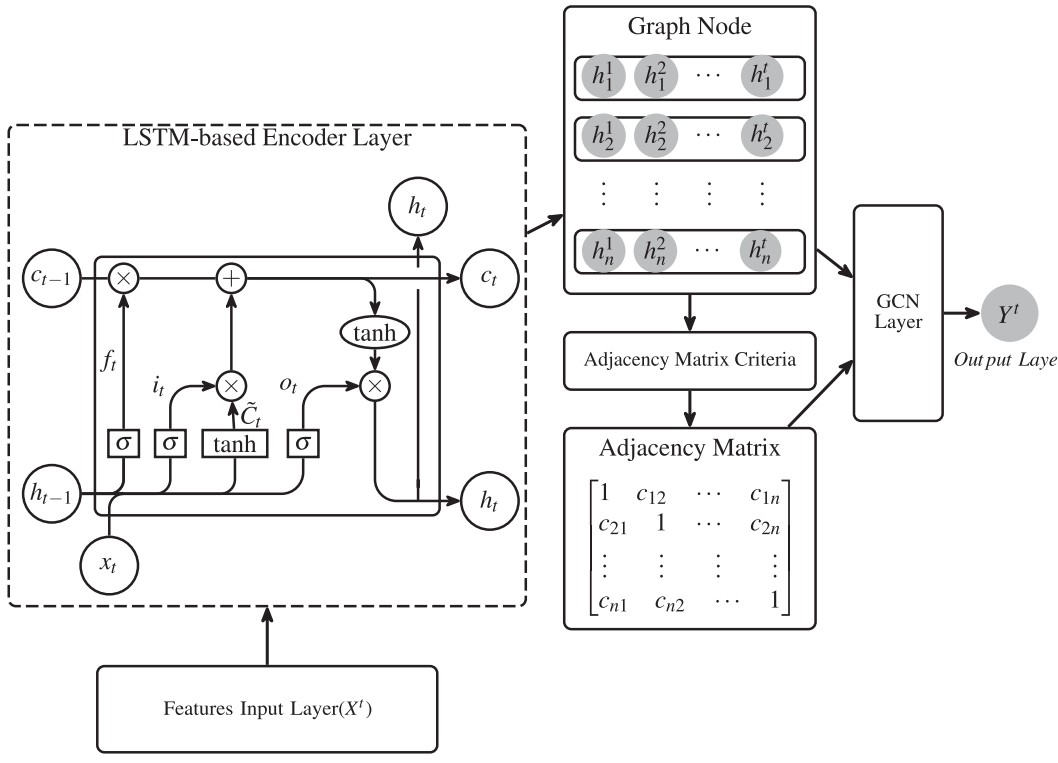

**Figure 1** A brief description of our proposed model.

construct graph data to achieve data fusion based on a comprehensive data set for stocks. LSTM and GRU are used to complete the aggregation process. An attention mechanism graph neural network based on nodes and edges is used to complete the classification and prediction.

## METHODS

In this section, we propose a novel model based on LSTM-GCN to extract stock features for stock trend prediction (Fig. 1). The proposed method consists of four parts: an LSTM network, a graph node creation along with its timing sequence feature, a Graph convolutional and a target prediction.

The proposed method is differ from the TSRM in *Zhao et al. (2023)*. In our model, all outputs of the hidden layers of sequences of the LSTM was utilized as the timing featres of the stock to better history time sequences. In the TSRM, the output of the hidden layer of the last sequence element of the LSTM was utilized as the timing feature of the stock. In our model, Pearson correlation coefficient (PCC) was used to quantify the stock relationship. In the TSRM, the authors used a K-means model to qualitative the stock relationship and the optional value of the stock relationship can only be 0 or 1.

## LSTM network

An LSTM network is used for extracting features of individual stock information. The LSTM network comprises three kinds of gating units and one cell, namely input gate $i_t$, output gate $o_t$, forgetting gate $f_t$ and memory cell state $c_t$. $t$ is a time step. The memory cell is like a conveyor belt. These gate units adaptively keep or override information into the memory cell. The function of $f_t$ is to control what information will be discarded from the memory cell. The function of $i_t$ is to control what new information will be stored into the memory cell. The function of $o_t$ is to select the information to be output from the memory cell (*Hochreiter & Schmidhuber, 1997*). The gates and the memory cell is calculated as in Eqs. (1)–(6).

$$i_t = \sigma(W_i \cdot [h_{t-1}; \ x_t] + b_i) \tag{1}$$

$$f_t = \sigma(W_f \cdot [h_{t-1}; \ x_t] + b_f) \tag{2}$$

$$g_t = tanh(W_c \cdot [h_{t-1}; \ x_t] + b_c) \tag{3}$$

$$c_t = f_t \odot c_{t-1} + i_t \odot g_t \tag{4}$$

$$o_t = \sigma(W_o \cdot [h_{t-1}; \ x_t] + b_o) \tag{5}$$

$$h_t = o_t \odot tanh(c_t) \tag{6}$$

where $W_i$, $W_f$, $W_o$, $W_c \in R^{d \times 2d}$ are the weight matrices of input gate, forget gate, output gate and memory cell, and $b_i$, $b_f$, $b_o$, $b_c \in R^d$ are the bias term of input gate, forget gate, output gate and memory cell. These weight matrices and bias term will be updated when minimizing loss function during LSTM training. $\sigma$ is an activation functions, $\odot$ is the element-wise multiplication. $h_t$ denotes a current hidden state. $h_{t-1}$ denotes a prior hidden state. $x_t$ denotes an input sequence.

## Graph node creation and its timing sequence feature

Stock market information includes individual stock data and the overall market topology structure. The topology structure can represented as a network graph where each graph node is a stock item $s_i$. The information of individual stock is transformed into an time sequence feature as the graph nodes of stock market network. The similarity of the feature of individual stock is transformed into the edge weight to construct the connections between graph nodes. Detail process of generating graph node embedding was presented in Fig. 2 and Eq. (7).

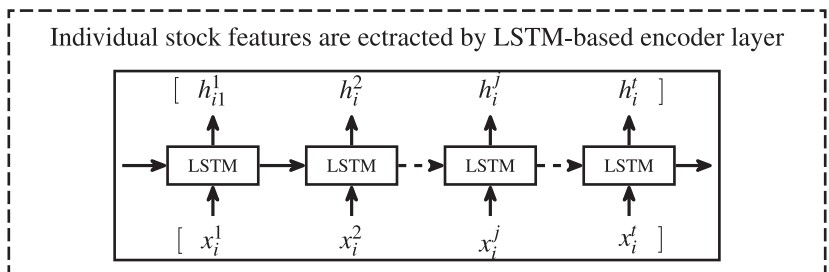

**Figure 2** **Process of generating graph node embedding for stock $s_i$ corresponding to graph node $g_i$.** Where, $x_i^t$ denotes the information of individual stock $s_i$ at timestep $t$. The number of the sampled information is $t$. The outputs of hidden state of LSTM network $h_i^j$ were utilized as the graph node embedding of stock $s_i$.

$$S^t : \begin{bmatrix} X_1^t \\ X_2^t \\ \vdots \\ X_i^t \\ \vdots \\ X_n^t \end{bmatrix} \xrightarrow{f} H^t : \begin{bmatrix} h_1^1 & h_1^2 & \cdots & h_1^t & p_1^{t+w} \\ h_2^1 & h_2^2 & \cdots & h_2^t & p_2^{t+w} \\ \vdots & \vdots & \vdots & \vdots & \vdots \\ h_i^1 & h_i^2 & \cdots & h_i^t & p_i^{t+w} \\ \vdots & \vdots & \cdots & \vdots & \vdots \\ h_n^1 & h_n^2 & \cdots & h_n^t & p_n^{t+w} \end{bmatrix} \tag{7}$$

In Eq. (7), $S^t$ denotes stocks information sampled at timestep $t$. $H^t$ denotes all output of hidden states at timestep $t$. $[p_1^{t+w}, p_2^{t+w}, \ldots, p_n^{t+w}]$ are the target price of stocks at next timestep $t + w$. As such, at timestep $t$ ($X_i^t$), we feed the sampled historical stock $i$ information to the LSTM network and take all hidden states $H_i^t$ as the embedding ($e_i^t$) of stock $i$ (notes that $e_i^t = H_i^t$). We have Eq. (8)

$$E_t = LSTM(X_t) \tag{8}$$

where $E^t = [e_1^t, \ldots, e_N^t]^{Transpose} \in R^{N \times U}$ denotes the sequential embeddings of all stocks. $U$ is the embedding size and the number of hidden units of LSTM. $N$ is the number of sampled stocks.

## Adjacency matrix representing the relationship

There are inter-stocks entity relationship, such as industry and upstream/downstream connections, and these relationship interact and influence each other (*Zhao et al., 2023*). In our work, PCC is used to measure these entity relationship. In statistics, PCC is also the most commonly correlation coefficient in financial analysis (*Chun-Xiao & Fu-Tie, 2018*; *Thakkar, Patel & Shah, 2021*). In *Chun-Xiao & Fu-Tie (2018)*, the authors used PCC to study the correlation of time series is to construct a correlation coefficient matrix. The authors found that the degree of a node is related to the average correlation coefficient between the node and other nodes. In *Thakkar, Patel & Shah (2021)*, the authors apply PCC for weight initialization instead of random initialization for a feed-forward vanilla neural network to enhance the prediction performance. The results demonstrate that the proposed weight initialization techniques provided comparable results compared to random initialization.

In this paper, PCC is used to construct adjacency matrix to handle adjacency relationship that exist among stocks. The PCC for variable $X$ and $Y$ can be obtained by using Eq. (9) (*Benesty et al., 2009*).

$$r = \frac{N \sum XY - \sum X \sum Y}{\sqrt{[N \sum X^2 - (\sum X)^2][N \sum Y^2 - (\sum Y)^2]}} \tag{9}$$

where $N$ denotes the number of stocks. $\sum$ is sum function. $X$ and $Y$ denote the sequential embeddings of all stocks. The explanation of $X$ and $Y$ can be referred to Fig. 2 and Eq. (7).

## Graph convolutional network

GCN is a special kind of graph-based learning methods to model the graph structure (*Kipf & Welling, 2017*), which is the state-of-the-art formulation for stock price prediction (*Chen, Wei & Huang, 2018*; *Feng et al., 2018*). In this paper, we use a simplified GCN (*Kipf & Welling, 2017*) to learn the relationship among stocks that is shown in Eq. (10). The GCN consists of two convolutional layers, one for input-to-hidden and the other for hidden-to-output. The GCN needs to be provided with two inputs: a matrix of node features $H$ representing the time-series features of stocks, a graph structure in the form of an adjacency matrix $A$ representing the inter-stocks relationship:

$$H^{(l+1)} = \sigma(\widehat{A} H^{(l)} W^{(l)}) \tag{10}$$

where $\widehat{A} = \check{D}^{-\frac{1}{2}} \check{A} \check{D}^{-\frac{1}{2}}$ is a normalized symmetric adjacency matrix. Here, $\check{A} = A + I$, $A$ is the adjacency matrix of the undirected graph G without added self-connections, $I$ is an identity matrix. $\check{D}$ is the degree matrix of $\check{A}$ and calculated by using the formula $\check{D}_{ii} = \sum_j \check{A}_{ii}$. $H^{(l)}$ is a $l$ layer-specific feature matrix and represents the hidden representations of all the nodes in the $l$-th layer. $H^{(l+1)}$ is the aggregated neighbor information for the $(l+1)$-th layer. $W^{(l)}$ is a $l$ layer-specific learnable weight matrix for the directed cross-correlation of stocks. $\sigma$ is a nonlinear activation function. In our case, we use $ReLU(\dots)$ as the activation function.

In this paper, we build a two-layer GCN model based LSTM network for stock price movements prediction. The graph convolutional of the GCN is described in Eq. (11):

$$Y = \sigma(\widehat{A} \sigma(\widehat{A} X W^{(0)}) W^{(1)}) \tag{11}$$

where $W^{(0)} \in R^{C \times H}$ is an input-to-hidden weight matrix, the hidden layer has feature matrix $H$. $W^{(1)} \in R^{H \times F}$ is a hidden-to-output weight matrix.

## Target prediction

The GCN output was used as the input for the output layer in order to make predictions of the stock's closing price on the following day. The output layer utilized in our model is a fully connected layer, as shown in Eq. (12):

$$\widehat{z}_i = W y_i + b \tag{12}$$

where $y_i$ is the output of stock $i$ from the GCN Layer. $\widehat{z}_i$ is the predicted close price of stock $i$ on the next day.

The proposed model was trained using smooth L1 loss (*Girshick, 2015*) over all stock data, because it is less sensitive to outliers than L2 loss. When the regression targets are
unbounded, training with L2 loss may require careful tuning of learning rates to prevent exploding gradients (*Girshick, 2015*). For a batch of size: math N, the unreduced loss can be described as Eqs. (13) and (14):

$$Loss = \sum \{l_1, \ldots, l_N\} \tag{13}$$

$$l_i = \begin{cases} 0.5(z_i - \widehat{z}_i)^2, & \text{if} |z_i - \widehat{z}_i| < 1 \\ |z_i - \widehat{z}_i| - 0.5, & \text{otherwise} \end{cases} \tag{14}$$

where, $z_i$ is the observed close price of stock $i$ on the next day.

## EXPERIMENTS

### Dataset description

Experimental data in this paper is from the daily trading of Chinese stock market. The constituent stocks of the MSCI China A 50 Connect Index (*MSCI, 2023*) were used as stock pool because the Index aims to reflect the performance of the 50 largest securities. The securities represent each Global Industry Classification Standard sector and reflect the sector weight allocation of the MSCI China A Index.[1] We defined an screening criteria of stocks that the stocks within the stock pool have uninterrupted data from 2018-01-02 to 2023-03-10 were selected as the experimental data. According to the screening criteria, 20 stocks from China Shanghai stock market were selected and 15 stocks from China Shenzhen stock market were selected. In this paper, the index change over days of the screened stocks is showed in Fig. 3. The computer method of the index is showed in Eq. (15), where i denotes the $i$th stock, the index was computed at the $t$-th day.

$$stock\ index^t = \sum_{i=1}^{35} \frac{colse\ price_i^t}{colse\ price_i^1}. \tag{15}$$

We use a publicly available API TdSdk (*TqSdk, 2023*) to get the historical daily quote data (including opening prices, high prices, low prices, closing prices, volumes etc.) of the screened stocks between 2nd January 2018 and 10th March 2023.

### Data preprocessing

We proposed a sliding window method (See Fig. 4), which was inspired by the sliding window method in *Chen et al. (2020)*. The proposed sliding window method, with a fixed start training time followed by a fixed length test period, effectively addresses the data alignment problem. The size of the training dataset started to steadily increase by increments of 44 days from 2nd January 2018, followed by a 44 days testing period in our experiments.

The data of the selected stocks is continuous from January 2, 2018 to March 1, 2023 without any interruptions, ensuring no issues with data alignment. The data values that do not fall within the stock opening period are removed. The quality of the dataset is excellent, with no errors or missing values.

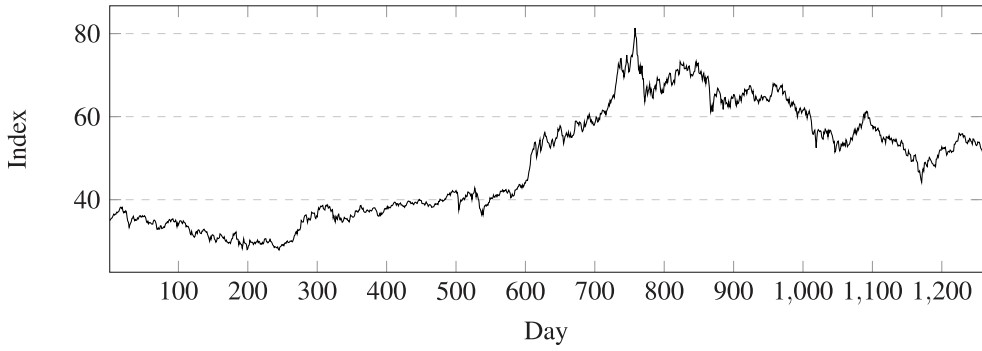

**Figure 3** **Daily chart of the stock A50 index from 2018-01-02 to 2023-03-10.**

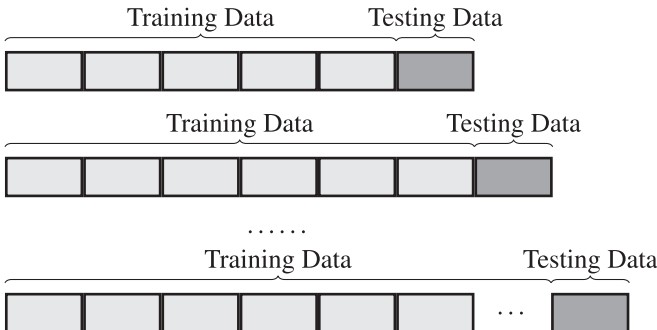

**Figure 4** **The proposed sliding window method, with a fixed start training time followed by a fixed length test period.** The training dataset is steadily increase by increments.

Normalization is a very good practice for the input data preprocessing to standardize the effect of price changes between different stocks. We rescaled the input sequences and target values to the range of 0 to 1 to avoid neuron saturation because the gates of the LSTM unit have an sigmoid activation function.

$$X_i^t = \frac{X_i^t}{max(X_i^t)*(1+0.1)-min(X_i^t)*(1-0.1)} \quad (16)$$

$$Z_i^{t+1} = \frac{Z_i^{t+1}}{max(X_i^t)*(1+0.1)-min(X_i^t)*(1-0.1)}. \quad (17)$$

In this paper, a new normalization method for input sequences and target values is proposed that is show in Eqs. (16)–(17). We increase the maximum value of each input data by 10% and decrease the minimum value by 10%, and then normalize the data to 0 to 1.

## Comparison methods

To show the performance and returns of the proposed model, we compare our model with some baseline methods, including an linear regression (LR), a naive Bayes (NB), a RF, a

Transformer based method, LSTM and GCN. The experiment was repeated five times for each method to minimize variability.

- **Linear regression**: this method takes the historical numerical information of stocks as the input data, applies linear logistic regression (https://scikit-learn.org/stable/modules/linear_model.html#logistic-regression) to predict stock price trend.
- **Naive Bayes**: this method takes the historical numerical information of stocks as the input data, and applies Naive Bayes classifier (https://scikit-learn.org/stable/modules/linear_model.html#bayesian-regression) to predict stock price trend.
- **Random Forest** (*Khaidem, Saha & Dey, 2016*): this method is an ensemble of multiple decision trees that minimizes forecasting error by treating the forecasting problem as a classification problem (https://scikit-learn.org/stable/modules/ensemble.html#forest). Some of the technical indicators such as relative strength index, stochastic oscillator etc were used as the inputs features.
- **Transformer based**: In *Zeng et al. (2023)*, the authors proposed a method for financial time series forecasting (CTTS) that based CNN and Transformer. Instead of the decoder of original Transformer (*Vaswani et al., 2017*), the proposed method used a multilayer perceptron with softmax activation.
- **LSTM** (*Nelson, Pereira & de Oliveira, 2017*): this method is a basic LSTM network. Based on the price history, the future trends of stock price were predicted alongside with technical analysis indicators.
- **GCN** (*Chen, Wei & Huang, 2018*): A three-layer GCN (*Kipf & Welling, 2017*) was used to incorporate information of related corporations of a target company for its stock price prediction.
- **TSRM** (*Zhao et al., 2023*): TSRM is an LSTM and GCN based model for stock price prediction, in which the stock relationships are measured by using a K-means model to cluster stock trading data, time series features are extracted by LSTM.

## Setting

We obtained the hyper-parameters for the proposed model based on validation set with AdaGrad algorithm (*Duchi, Hazan & Singer, 2011*), as shown in Table 1. Where, we look back 12 days to construct historical information. In particular, we use historical dat of the selected stocks from day $t-12$ to day $t$ as an input $X^t$, and predict the observed close price $Z^{t+1}$ on day $t+1$. $\widehat{Z^{t+1}}$ is the output of the proposed model that represents the predicted stock close price on day $t+1$.

The experimental environment was set as follows. The computer system was equipped with an Intel Core i5-12400 processor clocked at 2.50 GHz, 16 GB of RAM, a GeForce RTX 3070Ti graphics card, and Windows 11 operating system. The proposed model was implemented with Python 3.9.12, PyTorch 1.11.0 and CUDA 11.6.112. The DataParallel function of PyTorch 1.11.0 enabled parallel computation on the GPU. The integrated development environment utilized for the task was Visual Studio Code 1.73.1, which is provided by Microsoft.

**Table 1  Hyperparameter choices for the proposed model based on validation set.**

| Hyperparameter | Description | Value |
| --- | --- | --- |
| epochs | The number of epochs | 5 |
| lag | Time lag | 1 |
| input_len | The length of trading data features | 12 |
| input_size_1 | The number of expected features in the input | 2 |
| hidden_size_1 | The number of features in the hidden state | 128 |
| num_layers | The number of recurrent layers | 1 |
| input_size_2 | The number of input units | 1 |
| hidden_size_2 | The number of hidden units | 35 |
| out_features | The size of each output sample in the linear transformation to the incoming data | 1 |
| dropout | Introduces a 'Dropout' layer on the outputs of each LSTM layer except the last layer, with dropout probability | 0.2 |
| weight-decay | Weight decay (L2 penalty) in Adam algorithm | 5e−4 |
| lr | Learning rate in Adam algorithm | 0.01 |

## Evaluation metrics

The following metrics are adopted for prediction performance evaluation in the experiment: mean absolute error (MAE), mean square error (MSE), accuracy (Acc), precision (Pre) and recall (Rec) F-measure(F1) (*Chen et al., 2020*; *Zhao et al., 2022*; *Zhao et al., 2023*). The corresponding formulas are listed in Eqs. (18)–(26).

$$MAE = \frac{1}{MN}\sum_{i=1}^{M}\sum_{j=1}^{N}|z_{ij} - \hat{z_{ij}}| \tag{18}$$

$$MSE = \frac{1}{MN}\sum_{i=1}^{M}\sum_{j=1}^{N}(z_{ij} - \hat{z_{ij}})^2 \tag{19}$$

where, $M$ is the number of the screened stocks and $N$ is the number of samples in a stock. $z$ is the observed stock close price, and $\hat{z}$ is the output value of the proposed model.

$$Acc = \frac{TP + TN}{TP + FP} \tag{20}$$

$$Pre_{pos} = \frac{TP}{TP + FP} \tag{21}$$

$$Pre_{neg} = \frac{TN}{TN + FN} \tag{22}$$

$$Rec_{pos} = \frac{TP}{TP + FN} \tag{23}$$

$$Rec_{neg} = \frac{TN}{TN + FP} \tag{24}$$

$$F1_{pos} = \frac{2Pre_{pos}Rec_{pos}}{Pre_{pos} + Rec_{pos}} \tag{25}$$

$$F1_{neg} = \frac{2Pre_{neg}Rec_{neg}}{Pre_{neg} + Rec_{neg}} \tag{26}$$

where, true positive (TP) for correctly predicted event values, false positive (FP) for incorrectly predicted event values, true negative (TN) for correctly predicted no-event values, false negative (FN) for incorrectly predicted no-event values.

In stock trading backtest, the average annual return (AAR), sharp ratio (SR), maximum drawdown (MDD) and total returns money (TRMoney) metrics are used to focus on the trading risk and returns of the proposed model (*Chen et al., 2020*; *Zhao et al., 2022*; *Wang, Zhuang & Feng, 2022*). The corresponding formulas are showed in Eqs. (27)–(30). An annual risk free rate of 2.5% is adopted in the stock trading backtest.

$$TRMoney = Available\ Capital - Start\ Money + Closing\ Price \times Position\ of\ Stocks \tag{27}$$

$$AAR = [\frac{TRMoney}{Start\ Money}]^{1/Number\ of\ Year} \times 100\% \tag{28}$$

$$SR = \frac{Return - Risk\ freerate}{Standard\ Deviation\ of\ Return} \tag{29}$$

$$MDD = Max(\frac{P_x - P_y}{P_x}) \tag{30}$$

where, $P_x$ is the net value of stocks on day $x$, and $P_y$ is the net value of stocks on day $y$, $x$ is a certain day, $y$ is a certain day after $x$, annual risk free rate is 2.5%.

## Models training

The proposed model was trained end-to-end through error back-propagation algorithm. The training process was to minimize the loss function and adjust the model parameters step by step. AdaGrad algorithm was used to finished our training because its gradient-based adaptive optimization can regulate the learning rate of neural networks (*Duchi, Hazan & Singer, 2011*). AdaGrad algorithm is listed in Eqs. (31)–(33).

$$\theta_{t+1,i} = \theta_{t,i} - \frac{\eta}{\sqrt{G_{t,ii} + \varepsilon}} \cdot g_{t,i} \tag{31}$$

$$g_{t,i} = \nabla_\theta J(\theta_i) \tag{32}$$

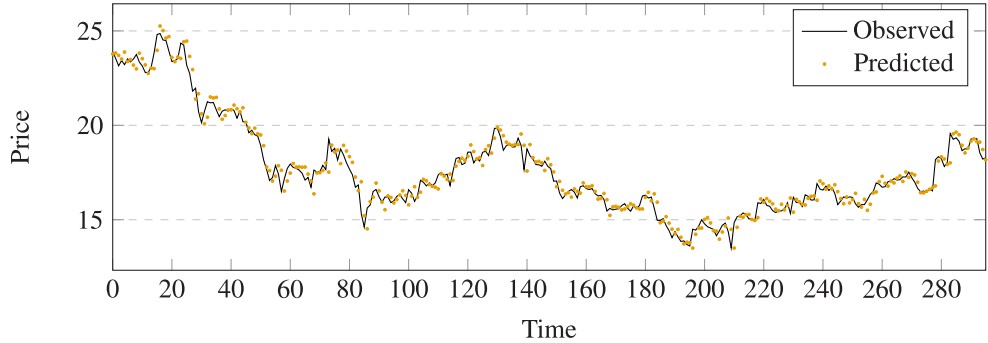

**Figure 5  Stock movement patterns of Linear Regression on stock 600031.**

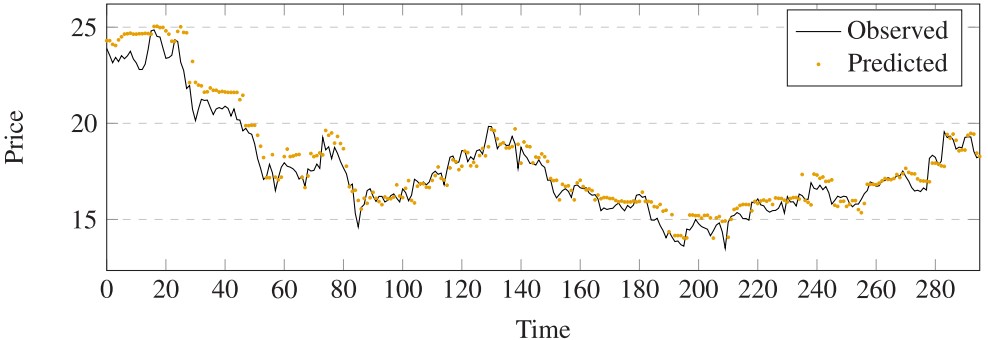

**Figure 6  Stock movement patterns of Navie Bayes on stock 600031.**

$$G_{t,ii} = \sum^{t} g_{t,i}^{2} \tag{33}$$

At time step $t$, the process to optimize parameter $\theta_{t+1,i}$ is performed according to Eq. (23). Where, learning rate $\eta$ is set to 0.01. $G_t \in R^{d \times d}$ is a diagonal matrix. Each element on the diagonal line $G_{t,ii}$ is an square sum over history gradient value $\theta_i$. $\varepsilon$ is a smoothing term that avoids division by zero (usually on the order of $1e - 8$).

## RESULTS

### Comparison results of different methods on stock 600031

The comparison results between prediction and output of different methods on stock 600031 are shown in Figs. 5–11. The prediction performance of the proposed model is more stable than baseline methods:

(1) As can be seen clearly from Figs. 5–11, LSTM-GCN (see Fig. 11) is more stable than other comparative methods.

(2) LSTM-GCN has no outliers unlike the GCN based method (see Fig. 9).

(3) LSTM-GCN has better continuity than LR (see Fig. 5), NB (see Fig. 6) and RF Regression based methods(see Fig. 7).

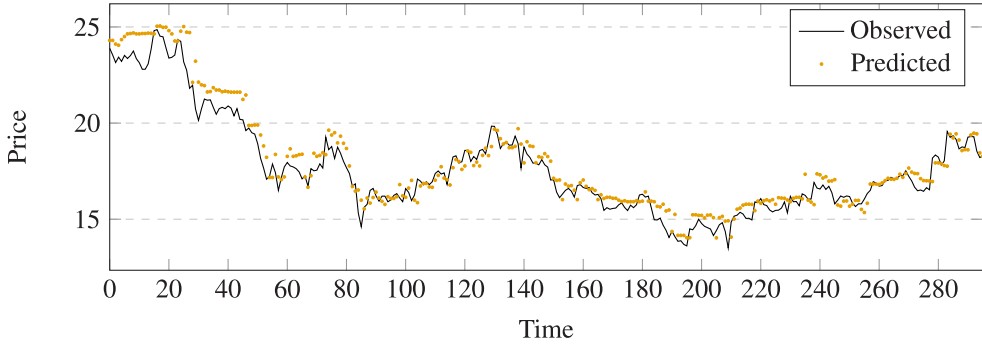

**Figure 7** Stock movement patterns of Random Forest on stock 600031.

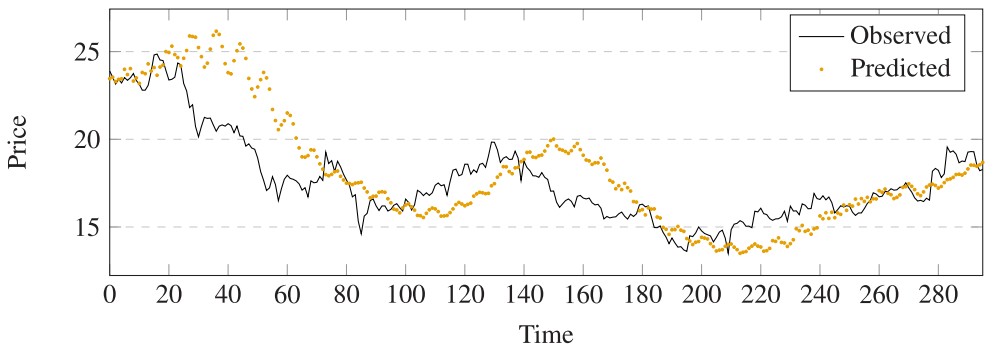

**Figure 8** Stock movement patterns of CTTS on stock 600031.

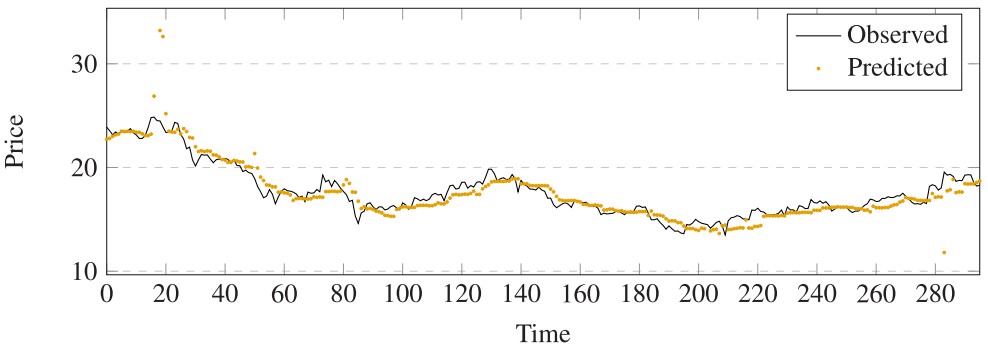

**Figure 9** Stock movement patterns of GCN on stock 600031.

(4) The difference between two values of LSTM-GCN is smalle all of CTTS (see Fig. 8), GCN (see Fig. 9) and LSTM (see Fig. 10) based methods.

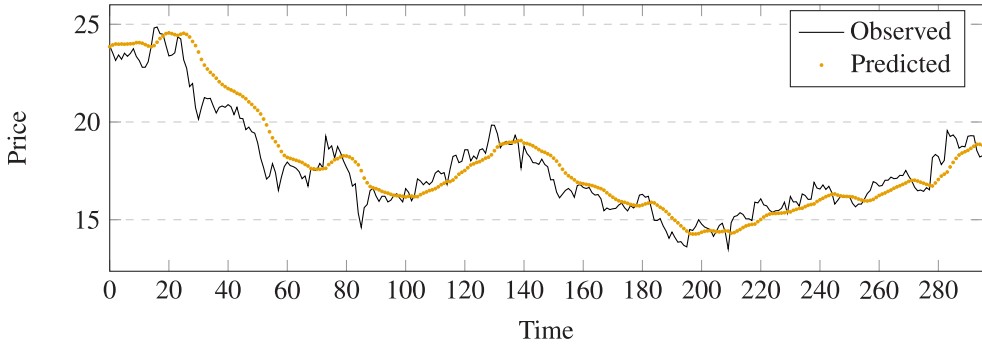

**Figure 10** Stock movement patterns of LSTM on stock 600031.

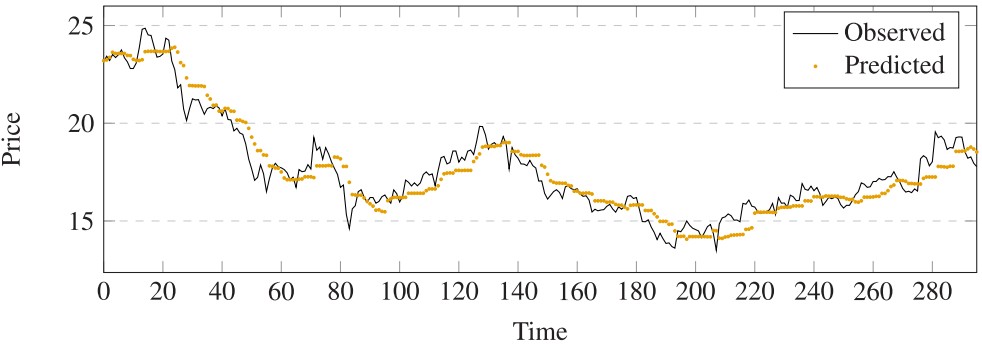

**Figure 11** Stock movement patterns of the proposed model on stock 600031.

**Table 2** The experiment results of our proposed model are compared with those of baseline methods based on stock 600031. The predicted price is at the next time step. The best results are shown in bold.

| code | Model | MAE | MSE | Acc(%) | $Pre_{pos}$(%) | $Pre_{neg}$(%) | $Rec_{pos}$(%) | $Rec_{neg}$(%) | 0 $F1_{pos}$(%) | $F1_{neg}$(%) |
|------|-------|-----|-----|--------|----------------|----------------|----------------|----------------|-----------------|----------------|
| 600031 | LR | 0.36 | 0.23 | 44.44 | 41.42 | 48.44 | 51.47 | 38.51 | 45.90 | 42.91 |
| | Naive Bayes | **0.35** | **0.22** | 46.46 | 43.43 | 50.82 | 55.88 | 38.51 | 48.87 | 43.82 |
| | RF | 0.56 | 0.52 | 46.46 | 44.60 | 51.19 | **69.85** | 26.71 | **54.44** | 35.10 |
| | CTTS | 1.47 | 3.88 | 50.34 | 46.54 | 54.74 | 54.41 | 46.88 | 50.17 | 50.51 |
| | LSTM | 0.69 | 0.83 | 50.34 | 46.67 | 54.96 | 56.62 | 45.00 | 51.16 | 49.48 |
| | GCN | 1.83 | 328.01 | 51.34 | 46.22 | 54.75 | 40.44 | **60.49** | 43.14 | **57.48** |
| | LSTM-GCN | 0.60 | 0.58 | **53.38** | **48.91** | **57.23** | 49.63 | 56.52 | 49.26 | 56.87 |

The evaluation metrics values of different methods on stock 600031 are list in Table 2. To better visualize the prediction performance of the proposed method, we have highlighted the best results in bold and created the bar charts based on these findings (See Fig. 12). The observation indicates that:

(1) LSTM-GCN outperforms all baseline methods in Acc, $Pre_{pos}$ and $Pre_{neg}$ metrics, remains medium values of $Rec_{pos}$, $Rec_{neg}$, $F1_{pos}$ and $F1_{neg}$ metrics, outperforms the deep

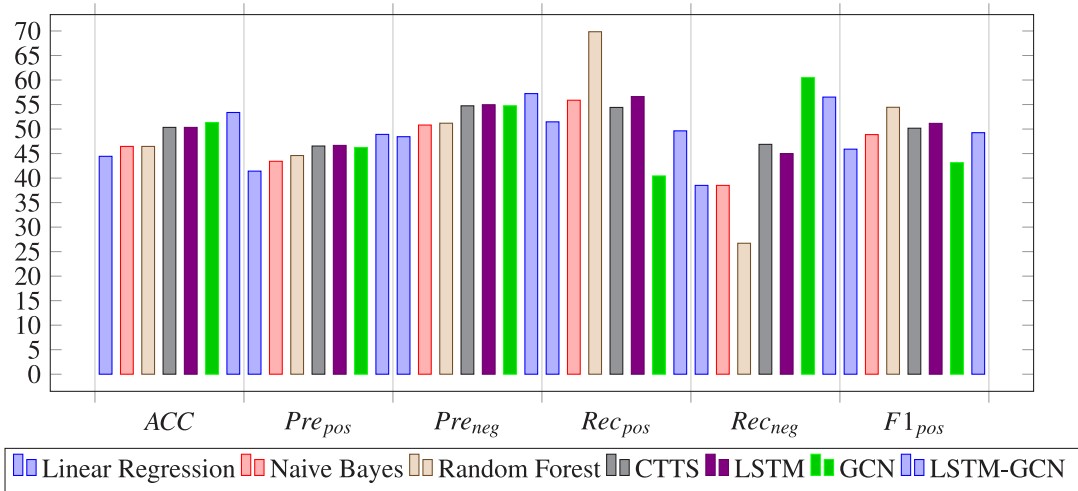

**Figure 12** The experiment results of our proposed model are compared with those of baseline methods based on stock 600031 by bar charts.

learning based methods (CTTS, GCN and LSTM) and is the same order as the machine learning based methods (LR, NB and RF) in MAE and MSE metrics.

(2) LSTM-GCN performs well than GCN and LSTM in MAE, MSE, Acc and Pre metrics, and remains medium values in Rec and F1 metrics. This indicates that the prediction performance can be improved by combining LSTM with GCN to integrate time series information and relational features of stocks.

(3) LSTM-GCN outperforms the deep learning based methods (CTTS and LSTM) in MAE, MSE, Acc, $Pre_{pos}$, $Pre_{neg}$ metrics due to its application on multiple stock input data.

(4) LSTM-GCN is the same order as the machine learning based methods (LR, NB and RF) in MAE and MSE metrics, and outperforms the machine learning based methods in Acc, $Pre_{pos}$, $Pre_{neg}$, $Rec_{neg}$, and $F1_{neg}$ metrics. This indicates that deep network structure is more effective for stock trend prediction.

(5) LSTM-GCN outperforms most baseline methods, but does not always stand out in all the evaluation metrics. It is clear that when predicting future trends of stock 600031, none of the methods always remain high values of MAE, MSE, Accuracy, Precision, Recall or F1.

## Comparison results of the proposed model and baseline methods on China A50 stocks

Table 3 shows the comparison results of the proposed model and baseline methods on China A50 stocks. We have highlighted the best results in bold. The conclusion derived in Table 3 is similar to those derived in Table 2. We drew bar charts based on Table 3 to clearly observe the prediction performance of LSTM-GCN (see Fig. 13).

**Table 3** **The experiment results of our proposed model are compared with those of baseline methods based on A50 stocks.** The predicted price is at the next time step. The best results are shown in bold.

| Model | MAE | MSE | Acc(%) | $Pre_{pos}$(%) | $Pre_{neg}$(%) | $Rec_{pos}$(%) | $Rec_{neg}$(%) | $F1_{pos}$(%) | $F1_{neg}$(%) |
|---|---|---|---|---|---|---|---|---|---|
| LR | 2.06 | 44.76 | 49.45 | 46.48 | 54.55 | 58.79 | 42.21 | 50.51 | 44.91 |
| Naive Bayes | **1.99** | **42.38** | 49.42 | 46.75 | 55.78 | 60.84 | 40.58 | **51.06** | 43.1 |
| RF | 7.05 | 277.26 | 48.42 | 44.3 | 52.86 | **65.06** | 35.43 | 49.84 | 36.28 |
| CTTS | 15.29 | 2226.03 | 50.06 | 46.34 | 48.82 | 59.97 | 42.78 | 47.42 | 39.37 |
| GCN | 5.32 | 1415.71 | 51.58 | 46.80 | 55.46 | 46.03 | **56.22** | 46.41 | **55.84** |
| LSTM | 6.15 | 240.32 | 50.31 | 43.78 | 55.1 | 54.1 | 47.86 | 42.62 | 43.39 |
| TSRM | 6.98 | 523.29 | 50.16 | 43.32 | 51.06 | 47.78 | 55.28 | 40.67 | 44.62 |
| LSTM-GCN | 3.18 | 102.84 | **51.69** | **47.40** | **56.47** | 54.82 | 49.07 | 50.84 | 52.51 |

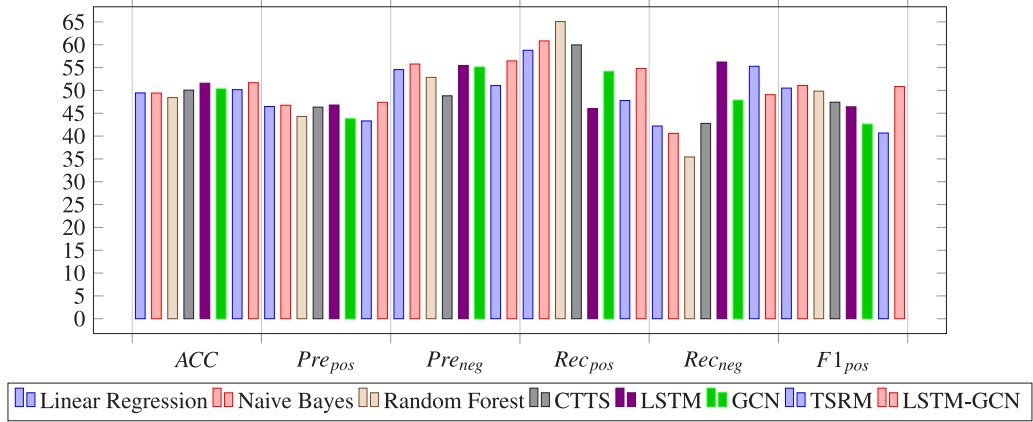

**Figure 13** **The experiment results of our proposed model are compared with those of baseline methods based on A50 stocks by bar charts.**

## Visualization of feature maps

The Figs. 14 and 15 display two maps that were obtained by registering a forward hook in the Implementation code of Grid-LSTM. The hook was called every time function 'forward' computed an output. Figure 14 visualizes a test input for LSTM-GCN, while Fig. 15 shows the corresponding feature extracted by LSTM-GCN. In Fig. 14, columns represent stock indices and rows represent time steps of stock data sampling. The visualization in Fig. 15 highlighting and distributing in a continuous manner compared to the input data in Fig. 14 demonstrates that information from the input data can be effectively extracted and centrally recorded. The experimental results based on stock market data from China further validate the effectiveness of these features.

## Stock trading backtest

We firstly identify a set of trading strategies as part of the trading plan to make more money than they lose. The strategies is following:

(1) The initial capital of each stock for trading was 300,000 RMB.

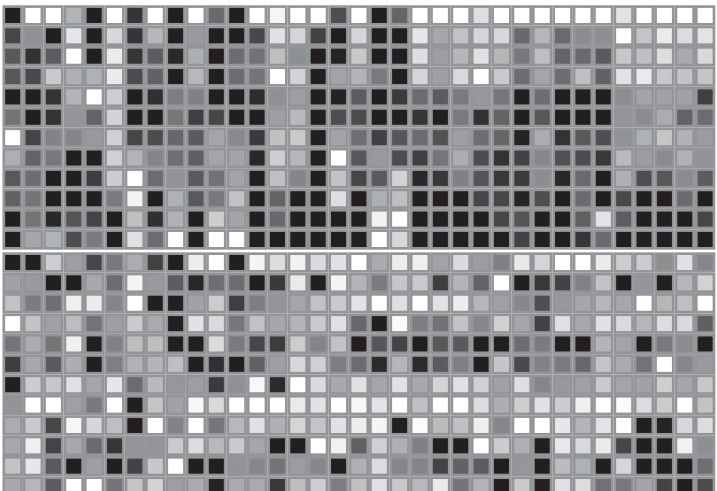

**Figure 14** The visualization of an input data from the test datasets.

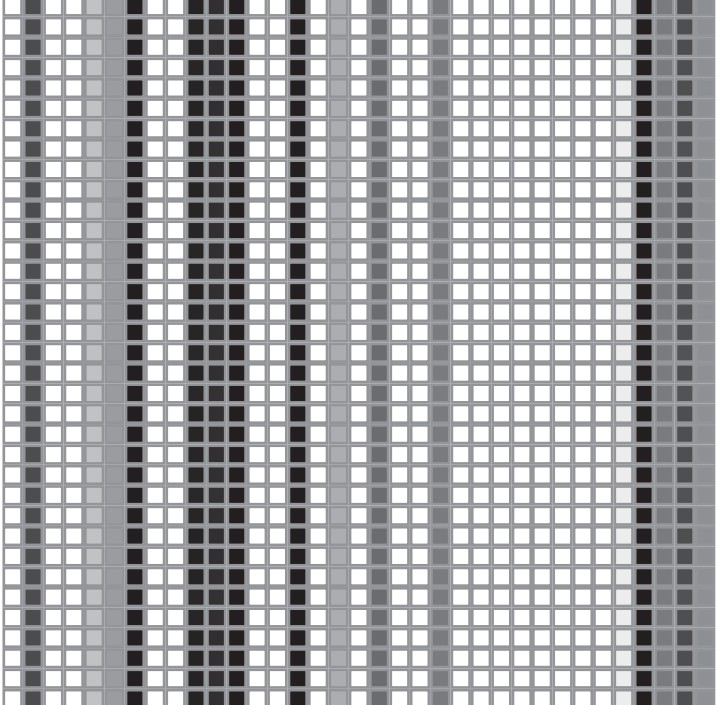

**Figure 15** The visualization of an extracted feature by LSTM-CCN based on the test datasets.

(2) The data from 2018-01-02 to 2023-03-10 was used for trading that was divided into many trading intervals. A trading interval is set to 44 days. The data before trading intervals is used for training.

(3) The predicted price is at the next time step.

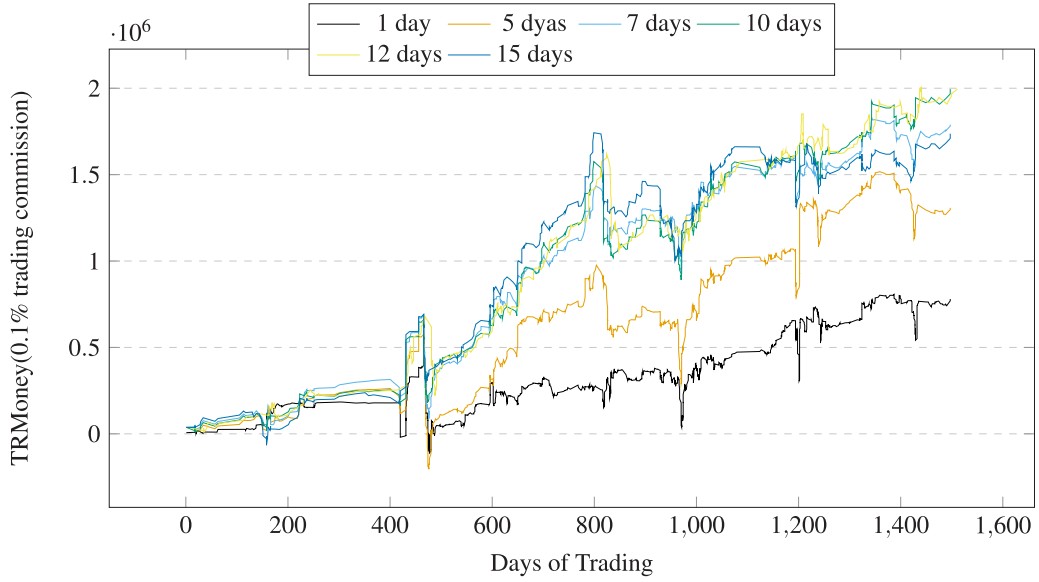

**Figure 16** Profits of A50 stocks using the various trading intervals over trading days (0.1% trading commission).

(4) Buy stocks when their forecast prices increase by 5% at next time step.

(5) Sell stocks when their profits exceed 5% at current time step to take profit from the curent stock trading.

(6) Sell stocks when holding them for more than 10 days at current time step. The predetermined trading intervals is employed instead of stop loss for the curent stock trading to prevent premature exit during stock liquidation. Figure 16 shows the choice of trading intervals and their impact on the results. The profits decrease when the trading intervals exceed 10 days or are less than 10 days. We can see that 10days is an appropriate trading intervals. The predetermined trading intervals are set to 10 days.

(7) Has no stock trading in other cases.

On financial evaluation, we train modes and simulated stock trading based on the screened stocks with using above strategies.

Table 4 shows the stock trading backtest results of LSTM-GCN and the baseline methods based on China A50 stocks. We have highlighted the best results in bold. We drew the profits changes over trading times (See Fig. 17) and the profits changes over trading days (See Fig. 18). The financial evaluation of LSTM-GCN is clearly observed. It is obvious that:

(1) LSTM-GCN has achieve more returns, has the best AAR and SR, medium level MDD than the comparative methods (see 4).

(2) The returns of all the methods are positive. AAR values of all the methods are more than 2%, SR values of five methods are more than 0 (see 4).

(3) LSTM-GCN can gain better returns regardless of the upward or downward range of the stock index (see Fig. 3), has good stability and robustness (see Figs. 17 and 18).

**Table 4  Comparison results for financial evaluation of our proposed model and baselines on A50 stocks.** TRMoney[1] Ignore trading commission, TRMoney[2] has an 0.1% trading commission. The best results are shown in bold.

| Methods | TRMoney[1] | TRMoney[2] | AAR | SR | Trading times | MDD |
|---|---|---|---|---|---|---|
| LR | 880,895.00 | 744,692.55 | 2.18% | −0.085 | 348 | 3.00% at 2021-1-25, times is 135 |
| Naive Bayes | 992,679.00 | 905,317.96 | 2.46% | −0.0138 | 222 | **2.18%** at 2021-2-4, times is 94 |
| RF | 1,917,202.00 | 1,610,046.18 | 4.99% | 0.515 | 782 | 3.68% at 2021-06-29, times is 348 |
| CTTS | 1,108,953.00 | 473,726.29 | 2.78% | 0.041 | 1629 | 10.56% at 2020-3-10, times is 547 |
| GCN | 1,827,452.00 | 1,554,276.52 | 4.73% | 0.490 | 695 | 5.55% at 2021-3-2, times is 253 |
| LSTM | 1,670,926.00 | 1,374,500.75 | 4.30% | 0.386 | 755 | 5.01% at 2019-4-4, times is 31 |
| TSRM | 1,311,444.00 | 877,463.50 | 3.32% | 0.153 | 1111 | 12.80% at 2019-4-11, times is 447 |
| LSTM-GCN | **2,270,575.00** | **1,994,924.60** | **6.00%** | **0.809** | 701 | 4.44% at 2021-3-1, times is 298 |

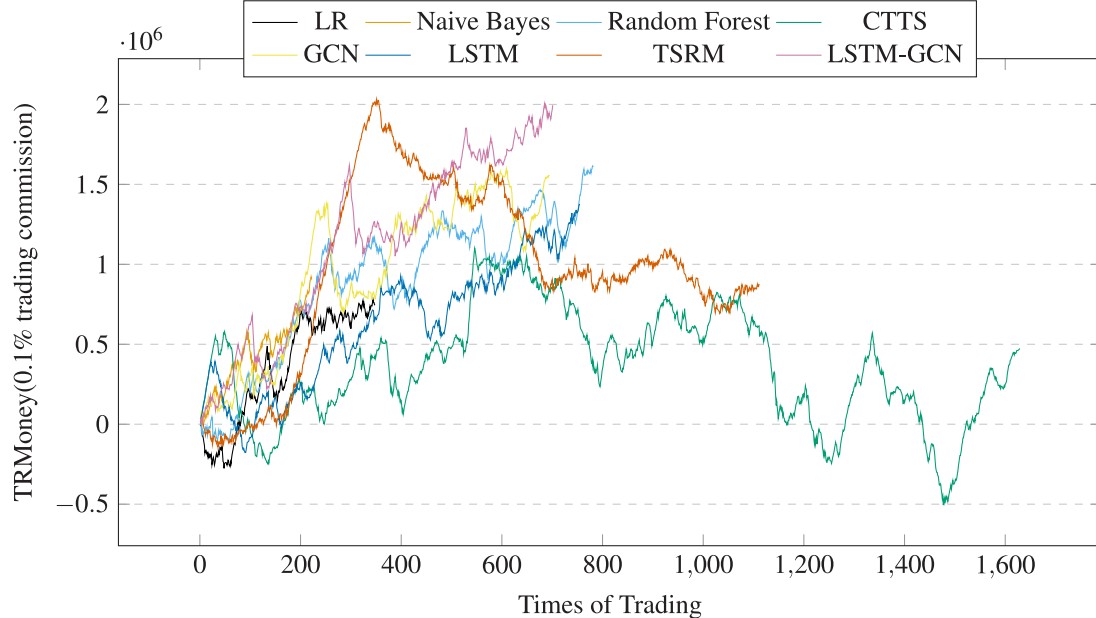

**Figure 17  Profits of A50 stocks over the trading times (0.1% trading commission).**

## LIMITATION

The study has potential limitations, as historical data may not always accurately predict future trends due to various trading circumstances and unforeseen market events that are not accounted for in the historical data. Incorporating hedging strategies into the model could be further explored in future research.

When applying the proposed model to live trading in the real stock market, it is crucial to adhere to practical constraints to avoid altering the supply and demand relationship and ensure the model's performance. For example, stock trading volume should not impact stock trends. Open position stock trading should be conducted using limit orders and dispersed trading, while close position stock trading should be conducted through dispersed trading.

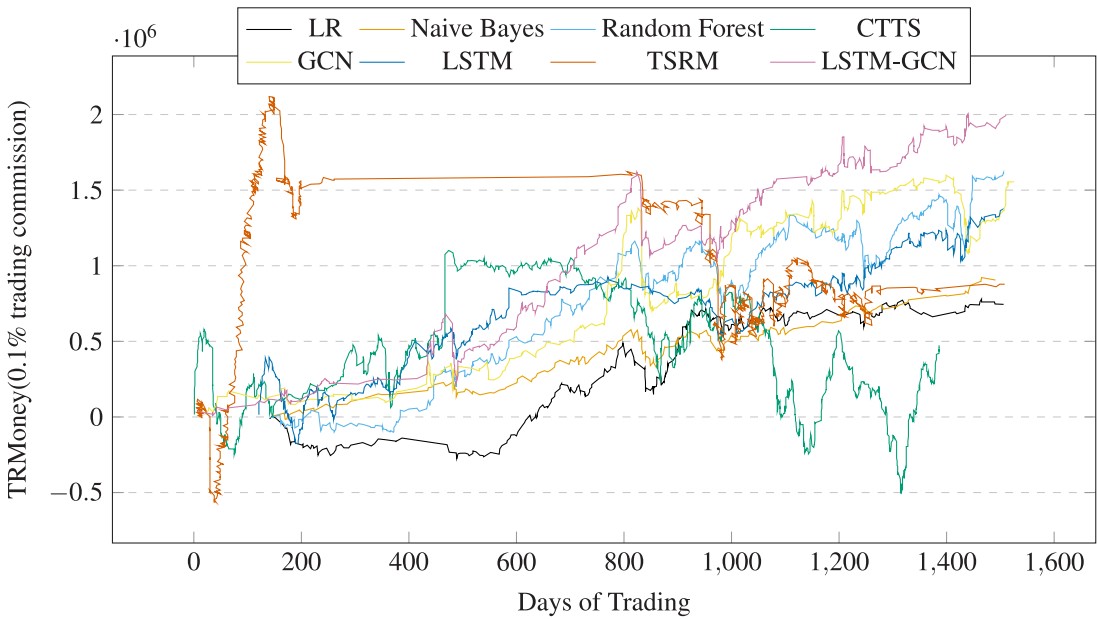

**Figure 18** Profits of A50 stocks over the trading days (0.1% trading commission).

## CONCLUSION

In this paper, we propose the LSTM-GCN model for stock trend prediction by modeling some dependencies between stocks that correlation and influence over stock price. We use an LSTM to extract the feature matrices from the corresponding stock, and all outputs of hidden state of the LSTM are utilized to create the graph node. The Pearson correlation coefficient is used to integrate the feature matrices to generate the adjacency graph matrix that represents the stock networks. Finally, the graph convolutional network is used to extract stocks features and the fully connected layer is used to predict price trends of target stocks. In trading backtest, we identify a set of trading strategies as part of the trading plan.

The experimental results demonstrate that the proposed model outperforms the baseline methods in terms of both prediction performance and trading backtest returns based on China A50 stocks. Other stocks or markets were not considered in this paper and can be part of future work. The outcomes of this research provide valuable insights for investors to make stock trading decisions: (1) By considering multiple stocks, the prediction performance and the trading backtest returns of the proposed model can be improved over baseline methods; (2) the proposed method achieve higher returns (BAR 6.00%), yielding a 20.2% improvement over the best baseline methods; (3) the proposed method demonstrates stability and robustness in financial evaluation, whether in upward or downward trends.

There are several ways in which our work could be extended. Some mechanisms need be added to adjust the model's predictions to handle sudden market changes or anomalies, such as financial crises or unexpected news events. For example, integrating text data such as financial news and social media posts that can be part of the reasons for stock price movement (*Li et al., 2023*), which may be beneficial both for the feature matrices or the

adjacency matrix. It is also straight forward to considering in combination with other dependencies to create adjacency matrix, such as ADM (*Zhu et al., 2022*), which may be better suited for modeling stocks relationships. Beyond stock trend prediction, LSTM-GCN can be generalized to other applications where graph network and trend prediction have been shown effective.

## ACKNOWLEDGEMENTS

The authors extend their appreciation to Mr. Congpeng Li for his interesting discussions on this work, and for generously providing the computing resources for model training.

### Funding

This research was funded by the College's Scientific Research Project of Beijing Information Technology College (number JB27132). The funders had no role in study design, data collection and analysis, decision to publish, or preparation of the manuscript.

### Grant Disclosures

The following grant information was disclosed by the authors:
The College's Scientific Research Project of Beijing Information Technology College: JB27132.

### Competing Interests

The authors declare there are no competing interests.

### Author Contributions

- Xiangdong Ran conceived and designed the experiments, performed the experiments, analyzed the data, performed the computation work, prepared figures and/or tables, authored or reviewed drafts of the article, and approved the final draft.
- Zhiguang Shan analyzed the data, authored or reviewed drafts of the article, and approved the final draft.
- Yukang Fan analyzed the data, performed the computation work, prepared figures and/or tables, and approved the final draft.
- Lei Gao analyzed the data, authored or reviewed drafts of the article, and approved the final draft.

### Data Availability

The data is available at GitHub: https://github.com/puma-running/LSTM-GCN.

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
