# Peer review of "A model based LSTM and graph convolutional network for stock trend prediction"

_PeerJ Computer Science, doi:10.7717/peerj-cs.2326_

## Round 0.1 · original submission · Major Revisions

· Academic Editor

Major Revisions

Dear authors,

The reviewers have commented on your paper. They indicated that it is not yet acceptable for publication in its present form.

However, if you feel that you can suitably address the reviewers' comments, I invite you to revise and resubmit your manuscript.

Please carefully address the issues raised in the comments.

If you are submitting a revised manuscript, please also:

a) outline each change made (point by point) as raised in the reviewer comments

b) provide a suitable rebuttal to each reviewer comment not addressed

Reviewer 1 ·

Basic reporting

The paper is well-structured and follows a logical flow, with clearly defined sections such as introduction, methodology, results, and conclusion. The figures and tables are relevant and presented, aiding in understanding the results. The language is primarily clear, but there are minor grammatical issues that need to be addressed for better readability (e.g., "smalle all of CTTS" should be "smaller than all of CTTS").

Experimental design

The experimental design is robust and well-explained. The authors used a comprehensive dataset covering stock 600031 and China A50 stocks. The use of multiple baseline methods for comparison is commendable, and the evaluation metrics are appropriate for assessing the performance of the models.
• A more detailed explanation of the data preprocessing steps would be beneficial.
• Clarification on the choice of trading intervals and their impact on the results.

Validity of the findings

The findings are valid and well-supported by the data presented. The proposed LSTM-GCN model shows superior performance compared to baseline methods across multiple metrics. The results are consistently illustrated through figures and tables, making the conclusions credible.

• A more detailed discussion on the study's limitations and potential biases in the dataset.
• Exploration of the reasons behind the model's performance in specific scenarios could be more detailed.

Additional comments

• Provide more details on the data preprocessing steps? how were missing values handled, and what techniques were used to normalize or standardize the data?
• What is the rationale behind the specific trading intervals for the stock predictions? How do different trading intervals impact the model's performance?
• Explain the choice of the LSTM-GCN model architecture in more detail? Why was this combination chosen over other potential hybrid models?
• How does the model handle sudden market changes or anomalies, such as financial crises or unexpected news events? Are there any mechanisms to adjust the model's predictions during such events?
• How do you ensure the generalizability of your model to other stocks or markets? Have you tested the model on different datasets to validate its robustness?
• Elaborate on the computational efficiency of the LSTM-GCN model? What are the resource requirements, and how does it compare to other models regarding training and prediction time?
• How do the results of the trading backtest translate to real-world trading scenarios? Are there any practical constraints or considerations that might affect the implementation of the model in a live trading environment?

Cite this review as

Reviewer 2 ·

Basic reporting

The authors proposed a model combining Long Short-Term Memory (LSTM) networks and graph convolutional networks (GCNs) to capture dependencies between stocks for improved stock trend prediction. Experiments on China A50 stocks demonstrated that their model outperforms baseline methods in prediction accuracy and trading backtest returns, identifying effective trading strategies and showing promising results in both rising and falling market conditions.


There is no mention of the study's limitations in the provided text. It is crucial to discuss the limitations of any research study to provide a comprehensive and balanced understanding of its strengths and weaknesses. By discussing the study's limitations, researchers can demonstrate transparency, acknowledge the potential shortcomings of their work, and encourage further research to refine and expand the knowledge in the field. Potential biases or risks associated with the model should be addressed. For example, reliance on historical data might not always predict future trends accurately due to unforeseen market events.

Experimental design

The manuscript should provide clear and detailed explanations about why LSTM and GCN were chosen, how the Pearson correlation coefficient is used, and the specific algorithms involved.

Validity of the findings

While the proposed approach demonstrates promising results, a comprehensive evaluation requires a comparison with existing studies that employ LSTM models for stock or index prediction. This comparison would provide valuable insights into the relative effectiveness of the proposed method. By examining the performance of the proposed model in comparison to previous works, researchers can gain a deeper understanding of its strengths and limitations. Additionally, such a comparison would contribute to the broader body of knowledge on LSTM-based stock or index prediction models, allowing for a more nuanced assessment of the proposed approach's contribution to the field.

Cite this review as

Reviewer 3 ·

Basic reporting

It is known that changes in stock prices in the same sector can follow a parallel course. However, although their dependencies cannot be observed directly and they are not in the same sector, some stocks can be quite synchronous in the short term. Research on how to use these relationships between stocks has been an interesting research topic in recent years.

In this article, LSTM was chosen to extract stock time series information and GCN was chosen to extract stock relationship information. Therefore, what is done in this article is actually to integrate stock relationships into stock price prediction. Thus, this integration is intended to enable the prediction of future stock prices and support investors in making investment decisions.

Although the integrated GCN-LSTM method is a method used in different fields, it has not been used much in stock price prediction. Zhao et al. (2023), which the authors also refer to in their study, is probably an exemplary study in this field in recent years.

In my opinion, the purpose, methodology and results of the authors' study should be compared with this literature in question. It should differ from them at some points and make itself unique. If necessary, the study should be revised again with new and original contributions for separation.

I expect convincing answers and satisfactory methodological revisions from the authors on these issues.

Experimental design

Although the integrated GCN-LSTM method is a method used in different fields, it has not been used much in stock price prediction. Zhao et al. (2023), which the authors also refer to in their study, is probably an exemplary study in this field in recent years.

Validity of the findings

The purpose, methodology and results of the authors' study should be compared with this literature in question. It should differ from them at some points and make itself unique. If necessary, the study should be revised again with new and original contributions for separation.

Cite this review as

---

## Round 0.2 · accepted · Accept

· Academic Editor

Accept

Dear authors,

Your revised version of the paper has been accepted by all reviewers. Contributions.

Reviewer 1 ·

Basic reporting

Fine

Experimental design

Good

Validity of the findings

Ok

Additional comments

Accepted
No further comments

Cite this review as

Reviewer 2 ·

Basic reporting

The manuscript is well prepared.

Experimental design

no comment

Validity of the findings

no comment

Additional comments

The authors made indicated changes in the previous revision.

Cite this review as

Reviewer 3 ·

Basic reporting

The literature, references, and context of the study seem to be sufficiently provided. The article structure, figures, and tables are placed appropriately. The analysis results seem to contain detailed evidence. My advice to the authors in future studies is to expand the scope in terms of data and test the methods once again in different market dynamics. Finally, I think an effort should be made to make the analyses that are difficult to understand for the reader simpler and more understandable. The depth and complexity of the analyses should not make one forget the main purpose and context. You should remember and emphasize the main benefit provided throughout the text.

Experimental design

It is seen that the research gap has been filled. The way the comparison with the literature was handled contributed to the study.

Validity of the findings

Although this study provides some insight into the validity of the current findings, different researchers should be encouraged to work on this methodology based on different market data in their future studies.

Additional comments

Although this study provides some insight into the validity of the current findings, different researchers should be encouraged to work on this methodology based on different market data in their future studies.

Cite this review as